# Muon measurements at the Pierre Auger Observatory

Dariusz Góra[1,*] for the Pierre Auger Collaboration [2,†,‡]

[1]Institute of Nucleatr Physics PAN, Radzikowskiego 152, Cracow
[2] Observatorio Pierre Auger, Av. San Martin Norte 304, 5613 Malargue
† auger_spokespersons@fnal.gov
‡ Full author list: http://www.auger.org/archive/authors_2022_07.html
* Dariusz.Gora@ifj.edu.pl

February 20, 2023

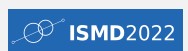

## Abstract

**The Pierre Auger Observatory is the world's largest detector for observation of ultra-high-energy cosmic rays (UHECRs) (above the energy of $10^{17}$ eV). It consists of a Fluorescence Detector (FD) and an array of particle detectors known as the Surface Detector (SD). Observations of extensive air showers by the Observatory can be used to probe hadronic interactions at high energy, in a kinematic and energy region inaccessible to experiments at man-made accelerators and to measure the muon component of the shower. Air showers induced by different primaries have different muon contents. With increasing mass of the primary cosmic ray particle, it is expected that the muon content in the corresponding air showers should also increase. Therefore, the determination of the muon component in the air shower is crucial to infer the mass of the primary particle. This is a key ingredient in the searches conducted to pinpoint the sources of UHECRs. Recent results obtained from the Pierre Auger Observatory and other experiments indicate that all the shower simulations underestimate the number of muons in the showers compared to the data. This is the so-called muon deficit. In this paper we briefly review the muon measurements, and present in more detail recent results on fluctuations in the muon number. These results provide new insights into the origin of the muon deficit in air shower simulations and constrain the models of hadronic interactions at ultrahigh energies. With the current design of the surface detectors it is also difficult to reliably separate the contributions of muons to the SD signal from the contributions of photons, electrons, and positrons. Therefore, we also present a new method to extract the muon component of the signal time traces recorded by each SD station using recurrent neural networks. The combination of such algorithms, with the future data collected by the upgraded Pierre Auger Observatory, will be a major step forward, as we are likely to achieve an unprecedented resolution in mass estimation on an event-by-event basis.**

## Introduction

Cosmic rays do not reach the Earth's surface directly but are subject to nuclear interactions in the atmosphere. Secondary particles produced as a result of such interactions undergo further interactions. These processes result in a cascade of particles, mainly electrons, photons and muons. At energies above $10^{16}$ eV the particles reach the Earth's surface, covering an area of up to several square kilometers. In this case, we are dealing with so-called extensive air showers (EAS). In order to describe how EAS are formed in the atmosphere, simple toy models, such as the ones described by Heitler and Matthews [1, 2], have been developed and are capable of providing accurate predictions of some of the quantities that characterize air showers without the need for high-performance computing. These models describe how, after the primary particle undergoes the first collision, each new interaction creates a new generation of particles until energy losses become too important and the cascade development ceases. Under certain approximations, the total number of particles of different types (electrons, photons, muons) found at each new generation, as well as the altitude at which the maximum of the shower is reached [1], can be simply expressed as a function of the primary energy and of the number of particles produced after the first collision. Although simplistic, the Heitler-Matthews model is powerful enough to allow the discrimination of EAS produced by proton/nuclei and photons. In particular, for hadronic showers, the dependence on the atomic mass $A$ of the primary cosmic ray energy $E$ highlights the fact that heavier nuclei produce showers with larger muon content. One of the reasons lies in the fact that lower energy nucleons transfer less energy into the electromagnetic component and more muons can be therefore produced. Moreover, showers initiated by heavier nuclei reach their maximum at shallower depth than in the case of primary protons or photons. The number of muons $N_\mu^A$ in an extensive air shower initiated by a nucleus with mass number $A$ can be related to the number of muons produced in a shower initiated by a proton with the same energy, $N_\mu^p$, through $N_\mu = N_\mu^p A^{1-\beta}$, where $1 - \beta \simeq 0.1$. Muons in EAS have also large decay lengths and small radiative energy losses and are produced at different stages of shower development. Therefore, muons can reach surface and underground detector arrays while keeping relevant information about the hadronic cascade.

In this context, it is worth noting that the recent results from studies of the muon component of large showers by leading high-energy experiments like IceCube, Pierre Auger Observatory, Telescope Array indicate that the currently used models of nuclear interactions at ultra-high energies significantly (at $\sim 8\sigma$ level) underestimate the production of muons in large atmospheric showers (the so-called muon deficit in simulations), see [3] for review of the different muon measurements. As an example, data from the Pierre Auger Observatory indicate that the observed number of muons is from 30% to even 60% larger than what we are able to reproduce in simulations of extensive air showers [4, 5]. Obviously, such a situation significantly affects the interpretation of experimental data, in particular the determination of the composition of cosmic rays. Therefore, studying the properties of nuclear interactions via measurement of muons and improving the ability to model the development of showers are also of great importance for understanding the properties and origin of cosmic rays.

---

[1]The atmospheric depth corresponding to this altitude is called the depth of shower maximum $X_{\mathrm{max}}$.

# Muon measurements

The Pierre Auger Observatory is a hybrid detector that combines a network of surface detectors with the four fluorescence sites with 27 telescopes which overlook the surface detector [6]. The FD can operate only during night hours with a low sky background and good weather conditions. The SD stations are water-Cherenkov detectors placed 1.5 km apart. Each station, powered by solar batteries, measures signals induced by the particles entering the water station. The observatory consists of 1660 SD stations, which form a giant network covering about 3000 km$^2$.

## Muon studies with inclined hybrid events

The electromagnetic component of EAS, which consists of electrons, positrons and photons, is mostly absorbed after the cascade travels about 2000 g/cm$^2$ through the atmosphere. This is the thickness of the atmosphere for a zenith angle of about 60°. As a result, the observed shower induced by a proton or a heavier nucleus at larger zenith angles contains mainly muons. Thus, by selecting the subsample of events reconstructed with both the SD and FD (the so-called hybrid events), and with zenith angles exceeding 60°, both the muon content and the energy of the shower are simultaneously measured [5].

In inclined showers, the number of muons is reconstructed by fitting a 2D model of the lateral profile of the muon density at the ground to the observed signals in the SD array [5]. The parametrized ground density for a proton shower simulated at $10^{19}$ eV with the hadronic interaction model QGSJetII-04 [8] was used. In such a case we can define $R_\mu$, the integrated number of muons at ground divided by the total number of muons $\langle N_\mu^{\mathrm{MC}} \rangle$ at the ground obtained by integrating the reference 2D model i.e. $R_\mu$ is the ratio between the integrated number of muons in the event and the total expected number of muons. Maximizing a likelihood considering detector response, physical fluctuations and the probability distribution of hybrid events we can get the $R_\mu$, see Figure 1 (left). The model used in the likelihood assumes that measurements of $E$ and $R_\mu$ follow Gaussian distributions centred at the true value, with widths given by the detector resolutions which defines the uncertainties obtained in each individual event reconstruction. Physical fluctuations are also assumed to follow a Gaussian distribution of width $\sigma$.

In Figure 1 (right) the average relative number of muons $\langle R_\mu \rangle$ from the likelihood fit is shown as a function of the measured energy. As we can see, the measurement does not fall within the range that is expected from high energy hadronic interaction models, even if hadronic interaction models commonly used to simulate EAS were updated to take into account LHC data at 7 TeV: QGSJETII-04 [8], EPOS-LHC [9] and SIBYLL-2.3c [10].

The average number of muons in a proton shower of energy $E$ has been shown in simulations to scale as: $\langle N_\mu^* \rangle = CE^\beta$ where $\beta \simeq 0.9$ [2, 11]. If we assume all the secondaries from the first interaction produce muons following the same relation as given for protons above, we obtain the number of muons in the shower as

$$N_\mu = \sum_{j=1}^{m} C \, E_j^\beta = \langle N_\mu^* \rangle \, \alpha_1 \tag{1}$$

where $\langle N_\mu^* \rangle = CE^\beta$ and $\alpha_1 = \sum_{j=1}^{m} (E_j/E)^\beta$ is the fraction of energy going into hadronic sector in the first interaction. The physical fluctuations for the next generation $i$ depend on multiplicity $m$ of secondary particles at each generation that interact hadronically and can be expressed by $\sigma(\alpha_i) \sim \frac{1}{\sqrt{m_1 \dot{m}_2 \ldots m_{i-1}}}$, see for details Ref. [12]. In other words, the $\alpha_2$ distribution for the second

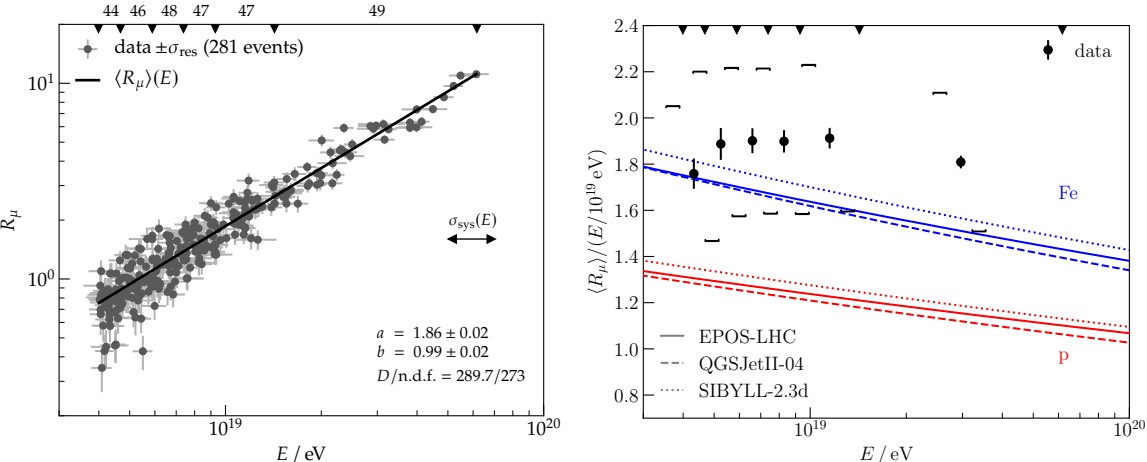

Figure 1: (Left) Relative number of muons in a shower as a function of the measured energy. The black line is the fitted $R_\mu = a\,[E/(10^{19}\,\text{eV})]^b$. Markers on the top of the frame define the bins in energy for which we will extract the average muon number $\langle R_\mu \rangle$ and shower-to-shower fluctuations $\sigma$, with the number of events in each bin shown above. The bins are chosen such that the number of events in each bin is similar. (Right) Measured average relative number of muons $\langle R_\mu \rangle$ as a function of the energy and the predictions from three interaction models for proton (red) and iron (blue) showers. Figures taken from [7].

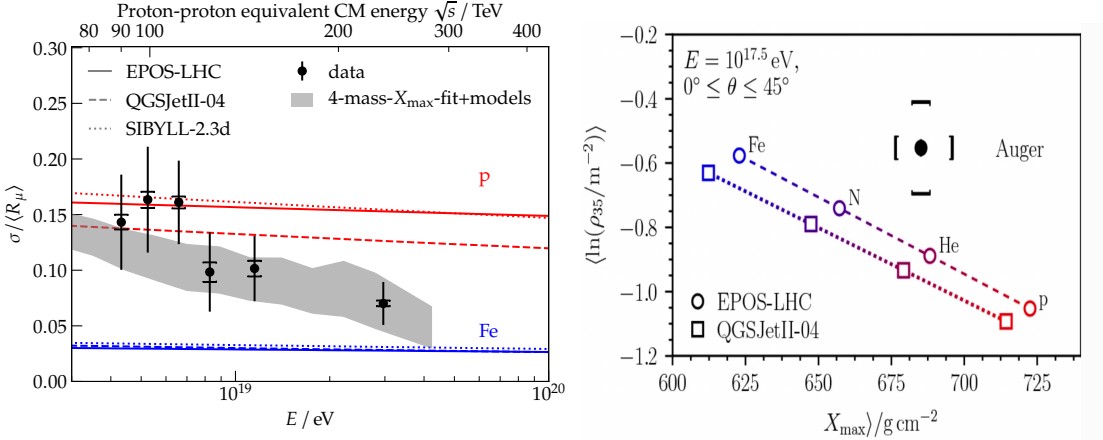

Figure 2: (Left) Measured relative fluctuations in the number of muons as a function of energy and the predictions from three interaction models for proton (red) and iron (blue) showers. The gray band represents the expectations from the measured mass composition interpreted with the interaction models. The statistical uncertainty in the measurement is represented by the error bars. The total systematic uncertainty is indicated by the square brackets. Figure taken from [7]. (Right) Mean logarithmic muon density $\ln(\rho_{35})$ as a function of the mean depth of shower maximum $X_{\text{max}}$ for simulations with primary energies of $10^{17.5}$ eV compared to UMD result. Figure taken from [13].

interactions is narrower by a factor $\sim 1/\sqrt{m_2}$. The deeper the generation $i$, the sharper the corresponding $\alpha_i$ is expected to be. As a result, the dominant part of the physical fluctuations will come from the first interaction.

In Figure 2 (left) we show measured relative fluctuations in the number of muons as a function of energy and the predictions from three interaction models for proton (red) and iron (blue) showers. The gray band represents the expectations from the measured mass composition interpreted with the interaction models. The muon relative fluctuations are in agreement with the mass composition expectations derived from the analysis of $X_{\max}$ data, suggesting that the muon deficit might be related to the description of low energy interactions.

Another import result of muon measurement comes from the underground muon detector (UMD) of the Pierre Auger Observatory. As part of the undergoing upgrade of Pierre Auger Observatory, the UMD is being deployed to obtain direct access to the muonic components in air showers produced by cosmic rays with energies between $10^{16.5}$ and $10^{18.5}$ eV with high statistics [13]. In the framework of the Heitler-Mathews model, the $X_{\max}$ and the muon number density measured in UMD - called here $\rho_{35}$- can be related to the mean logarithmic mass $\ln(A)$ through a linear dependence. Consequently, the relationship between $X_{\max}$ and $\ln(\rho_{35})$ can be represented by a line for each hadronic interaction model, as shown in Figure 2 (right) at energy $10^{17.5}$ eV. It is apparent that both models fail to reproduce the data. A difference of 38% in the muon number is observed compared to EPOS-LHC predictions, while the difference is larger compared to the QGSJetII-04 predictions. In both cases, data show that the analysed hadronic interaction models produce fewer muons than are observed in EAS. It is also worth noting that this is the first direct measurement of muon densities at energies $10^{17.3}$eV $< E < 10^{18.3}$ eV.

## Muon signals recorded with the Surface Detector using Recurrent Neural Networks

With the current design of the SD, the separation of the muon and electromagnetic components can only be done in events with large zenith angles or at distances far from the shower axis. Hence, for a majority of the data recorded by SD, this separation cannot be performed in a straightforward and efficient way. However, in Ref. [14] we perform an analysis which gives a possibility to estimate the muon signal for each station of SD by using a Recurent Neural Network (RNN) [15]. RNNs are specially well-suited for time-series due to their memory mechanism. There are several kinds of RNNs and, among them, we have chosen one of the most common, known as Long Short Term Memory (LSTM) [16, 17].

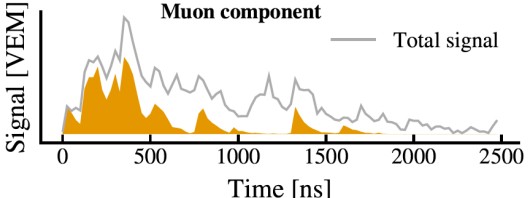 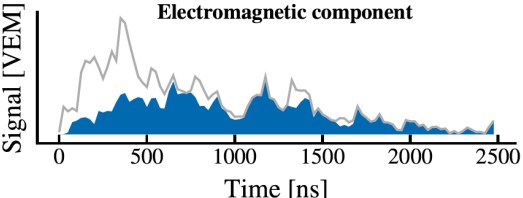

Figure 3: (Left) Muon component in an SD station from a simulated shower. Muons have earlier arrival times, usually spiky signals due to small attenuation and scattering of muons in the atmosphere. (Right) Electromagnetic component (signal from photons, electrons and positrons) in an SD station from a simulated shower. In this case we expect later arrival times, signal is spread in time but not very spiky, such traces are present mainly close to the shower axis.

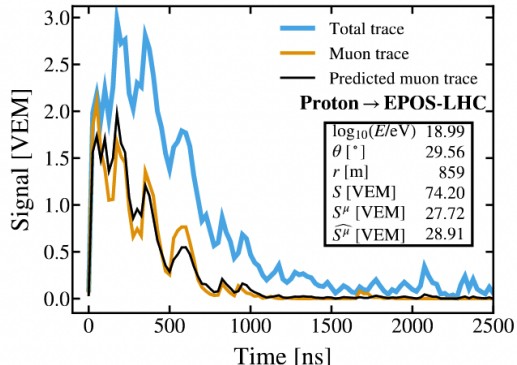 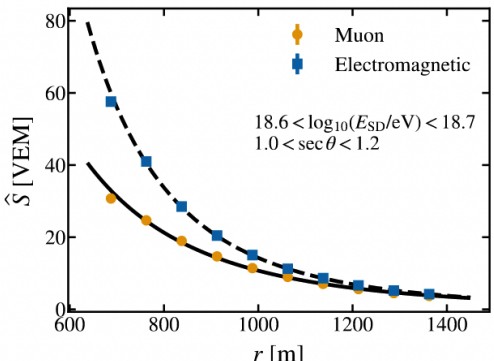

Figure 4: (Left) Examples of predicted muon trace for an event simulated with EPOS-LHC. The RNNs prediction (black line) agrees well with the shape of the simulated muon trace (orange line) for a majority of the time bins. The blue thicker line corresponds to the total trace, the one measured by an SD station. The integral of the muon signal is $S^\mu$, while the integral of the muon signal predicted by the network is $\widehat{S^\mu}$. (Right) The muon and electromagnetic signal from RNNs are fitted using functions obtained by the AGASA collaboration [18, 19], leaving only the normalization of the function free. The fits are in very good agreement with the signals predicted by the neural network from the measurements done by the Pierre Auger Observatory. Figures taken from [14].

The muons are highly penetrating particles – travelling along straight lines – therefore they usually arrive at the stations at earlier times than the electromagnetic particles. Furthermore, they produce a signal that is spiky, in contrast to the signal from electromagnetic particles which is more spread in time, see Figure 3. This is the basic physical principle for extraction of muon signal in an SD station by using RNN. The input for RNN is a time muon signal series of 200 time bins of the total signal measured by each of SD station. Moreover, the distance to the shower axis of each station $r$ and the secant of the zenith angle are also used in RNN training. The output is 200 time bins of the time trace. In Figure 4 (left) we show the results of such analysis. As we can see, the network successfully predicts not only the integral of the muon trace but also the shape of the muon trace. It reproduces the spiky shape and early arrival of muons. The predictions have a good resolution of 10%-20% of the total signal depending on the energy and zenith angle of the primary cosmic ray. This method works similarly well with other hadronic interaction models. When applied to data, the lateral distributions for the muonic and electromagnetic components follow the functional form obtained by the AGASA collaboration [18,19], see Figure 4 (right plot).

## Conclusion

The post-LHC hadronic interaction models are unable to provide a consistent description (except the muon relative fluctuations) of the showers recorded by the Pierre Auger Observatory over a wide energy range. The muon deficit is also experimentally established at $8\sigma$, by eight other leading air shower experiments [3].

Using a Recurrent Neural Network, the muon signal can be predicted for each water-Cherenkov detector of the Pierre Auger Observatory. The neural network is trained with simulations but the predictions are independent of the hadronic model used. The resolution of the integrals of

the predicted signals is between 10% and 20% of the total signal. Lateral distributions of muon and electromagnetic signals obtained with the RNNs from the Auger data agree well with the parametrizations obtained by AGASA [18, 19].

Determining the composition of cosmic rays based solely on the SD has been so far not as accurate as with FD. Therefore, the Auger Collaboration decided to enhance the capabilities of the surface array, which collects data continuously. This upgrade will enable more accurate measurements of the air shower particles at the ground level, in particular the separation of signals from the muonic and electromagnetic components on event-by-event basis, which is vital for cosmic ray mass composition studies. The combination of neural networks with the upgraded detectors of AugerPrime [20] will have an unprecedented performance regarding the estimation of the primary mass on an event-by-event basis.

# Acknowledgements

The successful installation, commissioning, and operation of the Pierre Auger Observatory would not have been possible without the strong commitment and effort from the technical and administrative staff in Malargüe. We are very grateful to the following agencies and organizations for financial support: https://www.auger.org/collaboration/funding-agencies. In particular we want to acknowledge support in Poland from National Science Centre grant No. 2016/23/B/ST9/01635, grant No.2020/39/B/ST9/01398 and from the Ministry of Science and Higher Education grant No.DIR/WK/2018/11 and grant No. 2022/WK/12 .

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
