# Peer review of "Muon measurements at the Pierre Auger Observatory"

_SciPost Physics Proceedings_

## Round 1 · Referee Report · Anonymous · 2022-12-12

Strengths
1. Relevant. This paper presents the status update on the issue of "muon deficit" observed in extended air showers.
2. Well written.
Weaknesses
1. Nothing obvious to me. The paper looks good.
Report
This paper is appropriate for publication in this proceedings. I have only one question and a minor suggestion to the text.
Requested changes
1. The definition of R_\mu is not completely clear to me. What's the difference between integrated number of muons and total number of muons? Is the total number independent of energy? I'm not suggesting a change to the text per se. But a clarification.
2. Page 2, 2nd paragraph: "60% larger what" -> "60% larger than what"
Author: Dariusz Gora on 2023-01-02 [id 3200]
(in reply to Report 1 on 2022-12-12)
Dear Reviewer,
Thanks very much for your comments, and please find below my answers:
-
Reviewer: "The definition of R_\mu is not completely clear to me. "
Answer: The R_\mu is the integrated number of muons at ground divided by the total number of muons N^{MC}{\mu} at the ground obtained by integrating the reference 2D model. R\mu= 1 in fact corresponds to N_\mu = 1.455 × 10^7 muons at the ground with energies above 0.3 GeV.
Reviewer:" What's the difference between integrated number of muons and total number of muons?"
Answer: In this context, the integrated number of muons N_{mu} is the value calculated for real event by using a maximum-likelihood method with the 2D reference model included in the fit, see for more details https://arxiv.org/pdf/1407.3214.pdf.
The total number of muons N^MC_\mu= ∫ dy ∫ ρ_{\mu,19} dx is obtained by integrating the reference 2D model, where ρ_{\mu,19} is the parametrized ground muon density i.e. a two dimensional function of position coordinates (x, y), for a proton shower simulated at 10^{19} eV with the hadronic interaction model QGSJetII-03. It was shown in detailed studies that the attenuation and shape of ρ_{\mu,19} depend very weakly on the cosmic-ray energy E and mass A for showers with zenith angle larger than 60 deg.
The number of muons N_\mu that have reached the ground is obtained and normalized to the number of muons N_{\mu,19} in the reference 2D distribution i.e. R_\mu= N_{mu}/N_{\mu,19}
Reviewer: "Is the total number independent of energy?"
Answer: As above, the total number of muons N^MC_{\mu} = ∫ dy ∫ ρ_{\mu,19} dx is obtained by integrating the 2D reference model. The attenuation and shape of muon density ρ_{\mu,19} depend very weakly on the cosmic-ray energy E and mass A for showers with zenith angle larger than 60 deg.
Reviewer: "I'm not suggesting a change to the text per se. But a clarification. "
-
Page 2, 2nd paragraph: "60% larger what" -> "60% larger than what"
This typo is now corrected in new version, see attached file D_Gora_ISMD-2022v2.pdf for this answer.
Best wishes Dariusz Gora
Attachment:
Anonymous on 2023-01-11 [id 3227]
(in reply to Dariusz Gora on 2023-01-02 [id 3200])Thank you very much for the clarification. I understand now. I'd suggest then something along these lines: R_mu is the ratio between the integrated number of muons in the event and the total expected number of muons.
Anonymous on 2022-12-12 [id 3127]
This paper is an update report of the issue of "muon deficit" for extensive air showers observed with the largest cosmic-ray detectors. The paper is well written and appropriate for publication in this proceedings. I have only one question and a minor suggestion. My question is about the definition of R_\mu. It's not clear to me the difference between the integrated number of muon and the total number of muons. Is the total number not a function of energy?
The minor suggestion is in page 2, second paragraph: I'd replace "60% larger what" with "60% larger than what".

---

## Editorial Decision

accepted_in_target_journal